# Dual-Double Stem Cell Ovarian Therapy: A Comprehensive Approach in Regenerative Medicine

**DOI:** 10.3390/ijms26010069

**Published:** 2024-12-25

**Authors:** Aleksandar Ljubić, Marija Dinić, Dajana Švraka, Svetlana Vujović

**Affiliations:** 1Pronatal Hospital, 11000 Belgrade, Serbia; dashinasuperhrana@gmail.com; 2Academy of Sciences and Arts of Bosnia and Herzegovina, 71000 Sarajevo, Bosnia and Herzegovina; 3Medigroup Health System, Dubrovnik International University, 20000 Dubrovnik, Croatia; 4Department of Therapeutic Apheresis, University Clinical Center of Serbia, 11000 Belgrade, Serbia; mediacentar@kcs.ac.rs; 5Clinic of Endocrinology, Diabetes and Diseases of National Center for Infertility and Endocrinology of Gender, 11000 Belgrade, Serbia; mf.bg@med.bg.ac.rs; 6Faculty of Medicine, University of Belgrade, 11000 Belgrade, Serbia

**Keywords:** dual-double stem cell regeneration, ovarian regeneration, MSCs, HSCs, ovarian decline

## Abstract

Dual-double stem cell therapy, which integrates mesenchymal stem cells (MSCs) and hematopoietic stem cells (HSCs), represents a cutting-edge approach in regenerative medicine, particularly for conditions such as ovarian decline, premature ovarian insufficiency (POI), and induced ovarian failure. This therapy leverages the unique properties of MSCs and HSCs, enhancing tissue repair, immune modulation, and overall regenerative outcomes. MSCs, known for their ability to differentiate into various cell types, provide a supportive microenvironment and secrete bioactive molecules that promote angiogenesis and reduce inflammation. HSCs, crucial for hematopoiesis and immune function, further enhance this environment by supporting hematopoietic processes and immune regulation. Clinical evidence increasingly supports the effectiveness of stem cell therapy in ovarian regeneration. Studies have demonstrated improved folliculogenesis, normalization of hormone profiles, and successful pregnancies in patients with POI. Furthermore, recent clinical trials in various medical fields underline the superior potential of dual-double therapy compared to monotherapies involving MSCs or HSCs alone, enhancing tissue repair and functional outcomes. However, despite these benefits, the therapy presents risks that require careful consideration. For autologous MSC therapy involving expanded cell populations, risks include tumorigenic potential, with evidence of sarcoma formation in certain cases of cultured MSCs. In contrast, autologous non-expanded MSC and HSC therapies may be limited by low cell yields, potentially compromising therapeutic efficacy. Additionally, non-expanded HSC therapy poses risks of insufficient cell numbers for successful engraftment and delayed immune reconstitution. These considerations underscore the importance of quality control and rigorous screening to optimize safety and efficacy. This article explores the mechanisms of action, clinical applications, and potential complications of dual-double stem cell therapy, underscoring the need for continued research and optimized protocols to enhance safety and outcomes in ovarian insufficiency and related conditions, offering new hope for affected women.

## 1. Introduction

Stem cell therapy has revolutionized regenerative medicine, offering novel treatments for various conditions. This article will discuss insights into the effects and mechanisms of stem cell therapy in medicine, specifically focusing on ovarian regeneration. It will outline the advantages of synergistic stem cell therapy combining MSCs and HSCs, along with potential risks and limitations.

The terms “dual” and “double” were included to reflect the flexibility of these two cell therapies, which can be administered simultaneously (dual) or sequentially (double). Combining MSCs and HSCs in dual-double stem cell therapy leverages the unique properties of each cell type to more effectively promote tissue repair and regeneration. This therapy has demonstrated promise across a range of clinical applications, including orthopedic injuries, cardiovascular diseases, and neurological disorders.

## 2. Types of Stem Cells Used in Therapy

Stem cell therapies offer promising regenerative and therapeutic possibilities across a variety of medical disciplines. Among the most studied stem cells are MSCs and HSCs, each with unique properties that contribute to their clinical utility.

MSCs are multipotent stromal cells capable of differentiating into osteoblasts, chondrocytes, myocytes, and adipocytes. They can be sourced from bone marrow, adipose tissue, umbilical cord blood, dental pulp, and placenta. MSCs are noted for their immunomodulatory and anti-inflammatory properties, allowing them to regulate immune responses and reduce inflammation, which makes them valuable in treating autoimmune diseases and chronic inflammatory conditions [1,2]. Additionally, MSCs secrete bioactive molecules—such as cytokines, growth factors, and exosomes—that support tissue repair, angiogenesis, and cell survival [3,4].

The regenerative properties of MSCs have led to their application across multiple fields. In orthopedics, MSCs are utilized for cartilage repair, osteoarthritis treatment, and bone healing [5,6]. In cardiovascular medicine, MSCs contribute to myocardial repair, reduce fibrosis, and improve cardiac function post-myocardial infarction [7]. The CHART-1 trial demonstrated the effectiveness of cardiopoiesis-guided MSC therapy in improving health-related quality of life in ischemic heart failure patients. Notably, the trial showed that MSC therapy significantly improved clinical outcomes, with reductions in mortality and hospitalization rates alongside enhanced patient-reported quality of life [7].

Investigations into MSCs’ potential to treat neurological conditions, including spinal cord injuries, stroke, and neurodegenerative diseases like Parkinson’s and Alzheimer’s, are also underway [8,9]. MSCs have demonstrated significant regenerative potential in treating neurological injuries. In animal models, MSC transplantation has led to notable functional recovery. For instance, a study by Quertainmont reported that rats with spinal cord injuries exhibited substantial improvements in motor function following MSC treatment, highlighting the cells’ capacity to promote neural repair and regeneration [10].

In clinical settings, MSCs have shown promise in enhancing recovery after stroke. A study by Kurniawan investigated the effects of human umbilical cord-derived MSCs on stroke patients [9]. The results indicated significant improvements in muscle strength, reduction in spasticity, and enhancement of fine motor functions, underscoring the neuroprotective and regenerative capabilities of MSC therapy in human subjects [11]. These findings suggest that MSCs offer a viable therapeutic approach for neurological conditions, promoting functional recovery through mechanisms such as neuroprotection, modulation of inflammation, and facilitation of tissue repair [12]. 

HSCs are primarily involved in hematopoiesis, the production of all blood cell types, including red blood cells, white blood cells, and platelets. They are found in bone marrow, peripheral blood, and umbilical cord blood. HSCs play an essential role in immune regulation, as they give rise to various immune cells, making them integral to bone marrow transplants and the treatment of hematologic diseases [13]. Clinically, HSCs are widely used in bone marrow transplantation for conditions such as leukemia, lymphoma, and severe combined immunodeficiency (SCID) [14]. Emerging research also highlights their potential in treating autoimmune disorders, including multiple sclerosis, systemic lupus erythematosus, and diabetes. HSCs are further utilized in managing blood disorders, such as anemias, thalassemia, and sickle cell disease, broadening their clinical impact [15].

## 3. Comparative Characteristics of MSCs, Adipose-Derived Stem Cells (ADSCs), Umbilical Cord Mesenchymal Stem Cells (UCMSCs), and HSCs

MSCs exhibit paracrine effects, secreting bioactive molecules like growth factors and cytokines that enhance their regenerative capacity [16].

Adipose-Derived Stem Cells (ADSCs) are obtained from adipose tissue, a rich source of MSCs and other regenerative cell types, including endothelial progenitor cells (EPCs) and pericytes [17]. Known for their high yield, ADSCs have strong angiogenic properties and robust paracrine effects, which promote vascularization and immune modulation, making them useful in regenerative medicine, aesthetic surgery, and cardiovascular treatment [18,19].

Umbilical Cord Mesenchymal Stem Cells (UCMSCs), sourced specifically from umbilical cords, exhibit high proliferation rates and low immunogenicity, making them suitable for allogeneic transplantation [20,21]. UCMSCs possess strong paracrine signaling abilities that support tissue repair and have shown promise in restoring ovarian function by regulating hormone levels and enhancing follicular development in cases of premature ovarian insufficiency (POI) [22,23].

HSCs are primarily applied in treating hematologic conditions, but recent studies indicate their potential role in ovarian recovery, particularly through systemic and paracrine effects that support follicular development and ovarian tissue repair [24,25].

## 4. Summary of Differences

The sources and yields of stem cells vary significantly between types. MSCs are obtained from multiple tissues, whereas ADSCs are specifically isolated from adipose tissue, which provides a high yield. UCMSCs are derived exclusively from the umbilical cord, while HSCs are sourced from bone marrow, peripheral blood, and umbilical cord blood.

UCMSCs demonstrate high proliferation rates and low immunogenicity, making them optimal for allogeneic applications. MSCs also exhibit proliferation and immunomodulatory properties, though these vary depending on the tissue origin.

Regarding differentiation potential, MSCs and ADSCs can differentiate into multiple cell types, directly supporting tissue regeneration. In contrast, HSCs are limited to hematopoiesis but indirectly aid tissue regeneration through immune modulation.

All types of stem cells exert paracrine effects, which promote tissue repair. Among these, MSCs and ADSCs notably contribute to angiogenesis and immune modulation, with ADSCs playing a prominent role in vascularization due to their endothelial progenitor cells (EPCs) content.

In terms of application in ovarian function, MSCs, UCMSCs, and ADSCs have demonstrated efficacy in ovarian restoration, particularly by enhancing folliculogenesis, improving hormone profiles, and supporting the ovarian microenvironment. Although HSCs are primarily utilized in hematologic therapies, they also support ovarian recovery when combined with other types of stem cells.

## 5. The Role of Stem Cells in Treating Ovarian Decline

Ovarian insufficiency, particularly premature ovarian insufficiency (POI), is a complex condition often arising from a combination of genetic, autoimmune, and environmental factors. While the exact etiology remains unclear in many cases, known causes include genetic mutations (such as in the FMR1 and FOXL2 genes), autoimmune disorders, and environmental factors like chemotherapy, radiation, and toxic exposures. Additionally, lifestyle factors, infections, and metabolic disorders may contribute to the early depletion of ovarian follicles, accelerating ovarian decline. These diverse etiologies complicate diagnosis and treatment, making recent advancements in stem cell research a promising therapeutic approach for affected patients.

## 6. Types of Stem Cell Therapy in Ovarian Regeneration

MSCs offer significant therapeutic effects in treating premature ovarian insufficiency (POI) through immunomodulatory, anti-inflammatory, and regenerative properties [26,27,28]. MSCs promote tissue repair by secreting bioactive molecules like VEGF, HGF, and IGF-1 [28,29,30]. These factors stimulate angiogenesis, increasing blood flow to ovarian tissue by approximately 30–40%, which supplies essential oxygen and nutrients. Additionally, MSCs help reduce apoptosis in ovarian cells, particularly granulosa cells, with studies indicating up to a 50% decrease in apoptosis levels [30,31]. Their immunomodulatory effect is also substantial; MSCs reduce inflammation by inhibiting pro-inflammatory T cells and natural killer cells, while promoting regulatory T cell formation, potentially leading to a 40–50% decrease in inflammatory markers within the ovarian microenvironment [31,32]. Together, these effects create a supportive environment for ovarian tissue regeneration and improved ovarian function in POI patients [33].

Studies in POI models have shown that MSCs can improve ovarian function by increasing follicle counts, enhancing hormone production, and reducing granulosa cell apoptosis. Some studies have shown a 30–40% increase in the number of primary and secondary follicles after MSC transplantation compared to untreated POI models [34].

MSC treatment has been associated with improvements in hormone levels critical for ovarian function, such as estrogen (E2) and anti-Müllerian hormone (AMH). Data from experimental models suggest that MSC treatment can lead to a 20–50% increase in estrogen levels and a significant restoration of AMH levels, indicating improved ovarian reserve [23,31]. MSCs are known to reduce apoptosis in ovarian granulosa cells, which are essential for follicle maturation. Quantitative data indicate that MSC treatment can lead to a 30–50% reduction in granulosa cell apoptosis, thus helping to preserve follicle health and functionality [35]. In addition to these specific effects, some studies measure the return of estrous cycles and even successful pregnancies in animal models, demonstrating the functional impact of MSC treatment. Additionally, clinical evidence supports MSCs’ efficacy, as some POI patients who underwent MSC transplantation regained menstrual cycles and fertility.

Wang et al. demonstrated that autologous bone marrow-derived MSCs could restore ovarian function in POI patients, with over 40% resuming menstrual cycles and some achieving spontaneous pregnancies, highlighting the potential of MSC therapy for fertility restoration [28]. Guo et al. further supported this approach, showing that MSC transplantation led to increased estradiol levels, follicle counts, and endometrial thickness. Approximately 30% of participants resumed menstruation, with a few reporting natural conception, indicating functional ovarian recovery [31].

Recent studies have investigated the potential of MSC therapy in treating premature ovarian insufficiency (POI), focusing on outcomes such as the resumption of menstruation, hormonal balance, and pregnancy rates. In a systematic review and meta-analysis [35], clinical trials involving MSC transplantation in POI patients were analyzed. The findings indicated that MSC therapy led to a significant decrease in serum follicle-stimulating hormone (FSH) levels (mean difference: −30.32 IU/L) and an increase in antral follicle count (AFC) (mean difference: 1.07). Additionally, the pregnancy rate improved by 19%, and the live birth rate increased by 19% among treated patients [35]. Another study by Herraiz et al. (2020) explored the effects of autologous bone marrow-derived stem cell transplantation in women with POI. The results demonstrated that 50% of the participants experienced the return of menstrual cycles post-treatment, and 25% achieved spontaneous pregnancies, underscoring the potential of MSC therapy in restoring ovarian function [25].

From a molecular standpoint, MSCs contribute significantly to ovarian regeneration through the secretion of key bioactive molecules, including VEGF, HGF, IGF-1, and bFGF. These molecules activate pathways such as PI3K/Akt and MAPK/ERK to promote angiogenesis, ensuring improved blood flow and nutrient delivery to damaged ovarian tissue. This vascularization supports cellular repair and follicular health. MSCs also exhibit strong anti-apoptotic and anti-inflammatory effects. Growth factors like VEGF, HGF, and IGF-1 inhibit granulosa cell apoptosis, which is essential for follicular development and ovarian function. MSC-derived cytokines and exosomes reduce inflammation by downregulating pro-inflammatory cytokines (e.g., TNF-α) and upregulating anti-inflammatory cytokines (e.g., IL-10), thereby protecting ovarian tissue from immune-mediated damage [36,37,38].

Additionally, MSCs prevent fibrosis by inhibiting excessive collagen deposition, maintaining tissue flexibility and a functional extracellular matrix (ECM) that supports follicular growth. Through these combined effects, MSCs create a supportive microenvironment that enhances ovarian recovery, positioning them as a valuable therapeutic approach for conditions like premature ovarian insufficiency (POI).

MSC-conditioned medium (MSC-CM) supports ovarian regeneration by delivering key bioactive molecules—VEGF, HGF, IGF-1, and bFGF—that activate cell signaling pathways essential for angiogenesis, cell survival, and tissue repair [39]. VEGF binds to receptors on ovarian endothelial cells, triggering PI3K/Akt and MAPK/ERK pathways that drive new blood vessel formation, supplying vital oxygen and nutrients to damaged ovarian tissues [40]. HGF interacts with the c-Met receptor, reducing apoptosis in granulosa cells and preserving follicular integrity, while IGF-1 enhances cell survival through PI3K/Akt signaling, supporting follicle health and proliferation [41]. bFGF contributes to extracellular matrix (ECM) remodeling, which is crucial for ovarian structural support and hormone signaling [42].

Adipose-derived MSCs (ADSCs) add anti-inflammatory and pro-angiogenic effects, secreting cytokines like IL-10 and TGF-β that decrease ovarian inflammation, creating a favorable environment for follicular survival [19,43]. Umbilical cord-derived MSCs (UCMSCs) aid in hormonal regulation through the hypothalamic-pituitary-ovarian (HPO) axis by stabilizing estradiol and AMH levels and supporting ECM integrity via matrix metalloproteinases (MMPs), promoting follicle growth [44,45]. Together, MSC-CM, ADSCs, and UCMSCs establish a regenerative microenvironment in ovarian tissue, combining angiogenesis, anti-apoptosis, hormonal stability, and ECM remodeling to enhance ovarian recovery and function, particularly in cases of premature ovarian insufficiency (POI) [46].

HSCs are primarily applied in hematologic treatments, but their potential in ovarian regeneration is gaining recognition. In cases of POI and chemotherapy-induced ovarian failure, HSCs contribute to ovarian recovery through systemic and paracrine effects, enhancing the ovarian microenvironment by secreting growth factors like VEGF, HGF, and IGF-1 [47]. These bioactive molecules promote angiogenesis, reduce inflammation, and inhibit apoptosis within ovarian tissue, thereby providing a supportive environment for follicle survival and overall ovarian function [48]. Moreover, the immunomodulatory effects of HSCs help mitigate chronic inflammation that may exacerbate ovarian damage, especially in patients undergoing chemotherapy [49].

Autologous stem cell ovarian transplantation (ASCOT) studies conducted by Herraiz et al. revealed that bone marrow-derived stem cells (BMDSCs) could restore ovarian function and promote follicle growth in POI patients [50]. The findings indicated that BMDSC infusion could lead to the development of preovulatory follicles and, in some cases, successful pregnancies, underscoring the potential of stem cell therapy in fertility restoration.

Another study by Herraiz et al. explored the effects of autologous bone marrow-derived stem cell transplantation in women with POI. The results demonstrated that 50% of the participants experienced the return of menstrual cycles post-treatment, and 25% achieved spontaneous pregnancies, underscoring the potential of MSC therapy in restoring ovarian function [50]. Similarly, research led by Pellicer et al. on BMDSCs showed improvements in hormone levels and follicular development in women with POI, further validating the role of stem cell therapy in addressing ovarian insufficiency [51]. These studies suggest that MSC transplantation may offer a promising therapeutic avenue for restoring ovarian function in POI patients. However, further research is necessary to confirm these results and establish standardized protocols for clinical application [52].

In an additional study, Ljubić et al. investigated the effects of autologous in vitro ovarian activation using bone marrow MSCs in combination with PRP growth factors, observing substantial improvements in endocrine and reproductive function in patients with ovarian insufficiency. This combination approach highlights the potential benefits of integrating stem cell therapy with growth factors for ovarian rejuvenation [53,54]. Research by Jadoul et al. also examined ovarian function in women treated with bone marrow transplantation during adolescence, identifying high-dose busulfan as a significant risk factor for ovarian failure. The study showed that patients experienced notable ovarian impairment, emphasizing the need for protective strategies in young women receiving chemotherapy [55].

A comparative analysis of different stem cell types—Mesenchymal Stem Cells (MSCs), Adipose-Derived Stem Cells (ADSCs), Umbilical Cord Mesenchymal Stem Cells (UCMSCs), and Hematopoietic Stem Cells (HSCs)—used in ovarian regenerative therapies can be seen in Table 1.

## 7. Advances in Stem Cell Therapy for Ovarian Function and Reproductive Outcomes

Emerging evidence suggests that stem cell therapy not only supports ovarian function but also improves reproductive outcomes [56]. For example, UCMSCs have demonstrated potential in restoring ovarian function and enabling pregnancies in both animal and clinical models [23,30]. Furthermore, MSCs and ADSCs have been shown to enhance folliculogenesis, leading to an increased number of developing follicles, which correlates with improved reproductive potential and fertility [57]. Additionally, new research underscores MSCs’ therapeutic effectiveness in ovarian regeneration for conditions such as POI and polycystic ovary syndrome (PCOS).

MSCs’ ability to enhance ovarian function through mechanisms, such as paracrine signaling, immune modulation, and direct tissue support, make them highly adaptable tools in regenerative medicine. Cell-free therapies with MSC-derived EVs offer promising, less invasive alternatives, expanding clinical applications for ovarian regeneration [58].

## 8. Dual-Double Stem Cell Therapy

Dual-double stem cell therapy leverages the combined effects of MSCs and HSCs, utilizing their synergistic properties for enhanced therapeutic outcomes. The application of dual-double stem cell therapy, which combines MSCs and HSCs, has shown promising outcomes across various medical conditions. In orthopedic conditions, MSCs effectively promote cartilage repair and reduce inflammation by differentiating into chondrocytes crucial for cartilage regeneration, while HSCs support bone marrow function and hematopoiesis, providing a continuous supply of cells necessary for healing. Pandey et al. demonstrated that autologous MSCs could alleviate knee pain and improve function in patients with osteoarthritis, with evidence of substantial cartilage growth and repair [59]. Current studies continue to support MSC therapy’s efficacy in reducing pain and enhancing joint function in orthopedic applications.

In cardiovascular medicine, MSCs and HSCs demonstrate significant potential for repairing damaged cardiac tissue and enhancing overall heart function, especially in patients with heart failure and myocardial infarction. MSCs can differentiate into cardiomyocytes and secrete bioactive molecules that support cardiac muscle repair, reduce fibrosis, and promote angiogenesis, crucially aiding in tissue repair and regeneration [60]. Additionally, HSCs contribute by enhancing blood flow through the promotion of neovascularization, which is critical in re-establishing oxygen and nutrient supply to ischemic heart tissues.

Based on findings from the ACCRUE meta-analysis and the DREAM-HF Phase 3 trial, cell therapy shows both potential and limitations in the treatment of cardiovascular diseases. The ACCRUE meta-analysis, which examined multiple studies of intracoronary cell therapy for acute myocardial infarction (AMI), found no significant improvements in key clinical outcomes such as left ventricular ejection fraction (LVEF) or infarct size reduction [61]. In contrast, the DREAM-HF Phase 3 trial, which investigated the use of allogeneic mesenchymal precursor cells (MPCs) in chronic heart failure with reduced ejection fraction (HFrEF), demonstrated promising results. Patients treated with MPCs showed improvements in left ventricular function and a reduction in major adverse cardiac events, including a decreased incidence of myocardial infarction and stroke, particularly among those with elevated inflammatory markers. These findings suggest that MPC therapy may provide anti-inflammatory and cardioprotective effects beneficial to patients with chronic heart failure [62].

In neurological applications, MSCs provide neuroprotection, promote neural regeneration, and modulate inflammation [63]. MSCs can differentiate into neural cells and secrete neurotrophic factors that support nerve growth and repair. HSCs contribute to neurogenesis and immune support, providing immune cells that can prevent secondary damage in cases of neural injury. Studies have shown significant functional recovery in rats with spinal cord injuries after treatment with MSCs, highlighting the regenerative potential of this therapy [9,64]. Recent investigations using umbilical cord MSCs in stroke patients demonstrated improvements in muscle strength, spasticity, and fine motor function, underscoring MSCs’ neuroprotective and regenerative capabilities [65].

## 9. Synergistic Effects of Mesenchymal and Hematopoietic Stem Cells in Ovarian Regeneration

The combination of MSCs and HSCs offers a comprehensive approach to ovarian regeneration, leveraging each cell type’s unique properties to enhance the regenerative process. The synergistic effect of MSCs and HSCs provides significant benefits for ovarian regeneration through enhanced paracrine signaling, immune modulation, and tissue repair.

Table 2 provides a comparative analysis of the key characteristics, roles, and regenerative functions of Mesenchymal Stem Cells (MSCs) and Hematopoietic Stem Cells (HSCs) in ovarian regeneration, highlighting their potential for addressing ovarian function decline and their synergistic therapeutic potential. 

Enhanced paracrine signaling from MSCs and HSCs supports ovarian regeneration through the secretion of growth factors and cytokines that activate specific molecular pathways [27,66]. MSCs secrete VEGF, HGF, and IGF-1, which bind to receptors on ovarian and endothelial cells, triggering the PI3K/Akt and MAPK/ERK pathways [67,68]. These pathways enhance angiogenesis, promote cell survival, and inhibit apoptosis, essential for revitalizing ovarian tissue. HGF, binding to c-Met, specifically protects granulosa cells, supporting follicular integrity and hormone production [69,70,71,72].

HSCs augment this environment by releasing IL-6, IL-10, TGF-β, and CXCL12, which modulate immune responses and recruit reparative cells. IL-10 and TGF-β provide anti-inflammatory effects, reducing tissue damage from chronic inflammation, a factor implicated in conditions like premature ovarian insufficiency. CXCL12 attracts additional reparative cells, reinforcing the regenerative niche. Together, the MSC–HSC combination amplifies paracrine signaling, fostering angiogenesis, immune modulation, and cellular survival, creating an optimal microenvironment for ovarian recovery and long-term function [73,74].

The immunological synergy between MSCs and HSCs in ovarian regeneration is achieved through their complementary roles in modulating immune responses at the molecular level. MSCs secrete anti-inflammatory molecules such as prostaglandin E2 (PGE2), transforming growth factor-beta (TGF-β), and indoleamine 2,3-dioxygenase (IDO), which collectively inhibit the activity of pro-inflammatory immune cells [75]. PGE2 reduces T cell activation and suppresses cytokines like IFN-γ and TNF-α, which can exacerbate ovarian damage. TGF-β further supports immune tolerance by expanding regulatory T cells (Tregs), essential for suppressing autoreactive T cells that could otherwise harm ovarian tissue [76,77]. IDO depletes tryptophan, crucial for T cell proliferation, effectively reducing T cell-mediated inflammation [76].

HSCs complement these effects by releasing cytokines like IL-10, an anti-inflammatory molecule that stabilizes the immune environment, and chemokines like CXCL12, which attract MSCs and other reparative cells to ovarian tissue [73]. Additionally, HSCs aid immune reconstitution by differentiating into immune cell lineages that restore immune balance, especially after chemotherapy. This combined action reduces ovarian apoptosis and fibrosis, creating a conducive environment for regeneration. Together, MSCs and HSCs establish an optimal immune landscape that protects ovarian cells, reduces inflammation, and promotes cellular recovery, facilitating ovarian tissue repair and function restoration [67].

The molecular synergy between MSCs and HSCs in ovarian regeneration involves a coordinated release of growth factors and cytokines that support tissue repair, differentiation, and cellular stability. MSCs secrete key factors like VEGF, HGF, and IGF-1, which activate pathways such as PI3K/Akt and MAPK/ERK, essential for promoting angiogenesis, cellular survival, and proliferation [78,79]. VEGF enhances blood supply to the ovarian tissue, while HGF protects granulosa cells from apoptosis through the c-Met receptor, preserving follicular structure [78]. MSCs also have the potential to differentiate into ovarian-like cells, directly replacing damaged cells. They release exosomes containing miRNAs that regulate gene expression for tissue repair, further supporting cellular recovery [79,80]. HSCs complement MSC actions by releasing cytokines such as IL-6 and chemokine CXCL12, which attract reparative cells to the site, enhance MSC recruitment, and stabilize differentiated cells within the ovarian tissue. This coordinated signaling creates an optimal environment for cellular repair, promoting both immediate recovery and long-term ovarian function.

Figure 1 presents a schematic depiction of the interaction between Hematopoietic Stem Cells (HSCs) and Mesenchymal Stem Cells (MSCs), emphasizing their cooperative roles in ovarian regeneration.

Ref. [25] conducted pivotal research demonstrating that the combined infusion of bone marrow-derived MSCs and HSCs could enhance ovarian function, increase follicle numbers, and restore hormonal balance in women with POI [25]. 

Notably, several patients achieved successful pregnancies following the treatment, highlighting the regenerative potential of this combined therapy [25].

## 10. Complications and Risks Associated with Autologous MSC and HSC Therapies in Ovarian Regeneration

Despite the promising potential of autologous MSC and HSC therapies for conditions like POI and chemotherapy-induced ovarian failure, there are notable complications that require careful consideration. In autologous MSC therapy, immune reactions may occur due to improper handling during isolation, expansion, or re-implantation, resulting in inflammation and adverse effects [81]. Additionally, MSCs’ high proliferative capacity and multilineage differentiation potential pose a risk of tumor formation under specific conditions, as evidenced by sarcoma formation in cases involving cultured MSCs [82,83,84]. The variability in MSC properties from different sources poses a challenge in maintaining consistent quality, impacting therapeutic efficacy.

Autologous HSC therapy carries risks associated with immune complications, especially if alterations in the recipient’s immune status occur post-transplant. The immunosuppressive treatments required to support HSC engraftment increase infection susceptibility, complicating recovery [85]. Conditioning regimens like chemotherapy or radiation, often necessary before HSC transplantation, can also result in significant ovarian damage, necessitating a balance to minimize ovarian harm while ensuring successful engraftment. Additionally, there is a risk of engraftment failure, where transplanted HSCs may not successfully repopulate the bone marrow [83].

## 11. Risks Associated with Autologous Non-Expanded MSC and HSC Therapies

The application of autologous non-expanded MSCs and HSCs introduces specific risks that must be considered carefully. Non-expanded MSC therapy may be limited by the low cell quantity available, potentially compromising therapeutic efficacy, as higher cell doses are often necessary to achieve significant effects. Quality control challenges arise as non-expanded MSCs, directly harvested from the patient, can vary significantly in potency and regenerative capability. Infection risk is heightened due to the lack of expansion procedures, emphasizing the importance of aseptic techniques to prevent severe complications [84].

Autologous non-expanded HSC therapy may result in insufficient cell numbers for successful engraftment, leading to suboptimal clinical outcomes. There is also a risk of reintroducing residual disease in patients who have undergone chemotherapy, necessitating thorough screening to confirm the absence of malignant cells in harvested HSCs. Lastly, non-expanded HSCs may delay immune reconstitution, increasing the risk of infections and immune-related complications [85].

## 12. Conclusions

The combination of MSCs and HSCs offers a comprehensive approach to ovarian regeneration by harnessing the unique regenerative and immunomodulatory properties of each cell type. Dual-double stem cell therapy has demonstrated significant potential for restoring ovarian function and enhancing reproductive outcomes. However, it is essential to carefully consider the associated risks, including immunogenicity, tumorigenicity, infection susceptibility, and complex immune interactions. Ongoing research, carefully controlled clinical trials, and extensive patient monitoring are essential to optimize therapeutic outcomes and ensure patient safety. As regenerative medicine evolves, dual-double stem cell therapy continues to be a promising frontier for ovarian regeneration and broader applications in regenerative health.

## 13. Take-Home Messages

The combination of MSCs and HSCs offers a synergistic approach to ovarian regeneration by harnessing the unique properties of each cell type. This dual-double stem cell therapy promotes tissue repair, immune modulation, and enhanced regenerative outcomes, making it particularly promising for conditions like premature ovarian insufficiency (POI), ovarian decline, and therapy-induced ovarian failure. Clinical evidence supports the therapy’s effectiveness in improving ovarian function, restoring hormonal balance, and potentially enhancing fertility outcomes.

However, the application of dual-double stem cell therapy is not without risks. Potential complications, such as immunogenicity, tumorigenicity, and intricate immune interactions, necessitate careful monitoring and management to ensure patient safety. Continued research and clinical trials are essential for optimizing both the safety and efficacy of combined MSC and HSC therapies, ensuring that the therapeutic benefits justify any associated risks.

Moreover, the variability in stem cell sources and individual patient factors underscores the importance of a personalized medicine approach. Tailoring treatment plans to the specific needs of each patient is vital for achieving the best possible outcomes in regenerative medicine, especially within the nuanced field of ovarian therapy.


## Figures and Tables

**Figure 1 ijms-26-00069-f001:**
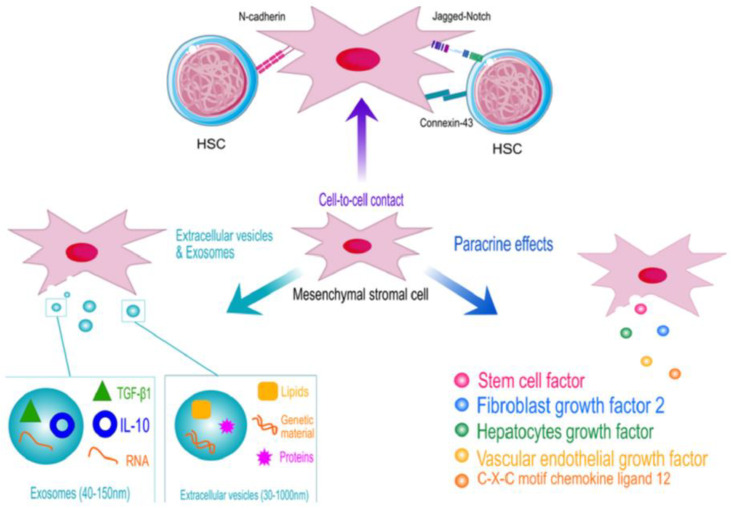
Depicts a schematic representation of the interaction between hematopoietic and mesechymal stem cells.

**Table 1 ijms-26-00069-t001:** Provides a comparative overview of different stem cell types—Mesenchymal Stem Cells (MSC), Adipose-Derived Stem Cells (ADSC), Umbilical Cord Mesenchymal Stem Cells (UCMSC), and Hematopoietic Stem Cells (HSC)—used in ovarian regenerative therapies. Each cell type has unique properties and potential applications in supporting ovarian regeneration and addressing ovarian function decline.

Cell Type	Source	Key Characteristics	Application in Ovarian Regeneration
MSC (Mesenchymal Stem Cells)	Bone marrow, adipose tissue, umbilical cord	Multipotent, immunomodulatory, secrete factors [1,2,16]	Improve folliculogenesis and hormone levels in POI [4,23,25]
ADSC (Adipose-Derived Stem Cells)	Adipose tissue	High proliferative capacity, promotes angiogenesis [17,18,19]	Enhance vascularization and support ovarian microenvironment [26,28]
UCMSC (Umbilical Cord MSC)	Umbilical cord	High proliferation rate, low immunogenicity [20,21,22]	Regulate hormone levels and increase viable follicles in POI [22,23,30]
HSC (Hematopoietic Stem Cells)	Bone marrow, peripheral blood	Supports hematopoiesis, immune modulation [11,12,13]	Assist in ovarian function restoration by enhancing ovarian microenvironment and supporting tissue repair through immune signals [24,25]

**Table 2 ijms-26-00069-t002:** This table presents the characteristics of MSCs and HSCs relevant to their potential effects on ovarian regeneration. It compares the primary roles and regenerative functions of MSCs and HSCs in maintaining ovarian health, with a focus on their potential to address ovarian function decline. The synergistic use of these cells offers a promising therapeutic approach for ovarian regeneration.

Cell Type	Key Characteristics	Functions in Ovarian Regeneration	Clinical Applications
Mesenchymal Stem Cells (MSC)	Multipotent, secrete bioactive molecules; can differentiate into bone, cartilage, fat cells; sourced from bone marrow, adipose tissue, umbilical cord blood, etc. [16,17,18,20].	Promoteangiogenesis via VEGF and bFGF [4,19,23]; -Reduce inflammation (IL-10, TGF-β) [2,16]; -Inhibit apoptosis in granulosa cells to preserve follicles [23].	Enhances folliculogenesis, hormone levels, and ovarian function in cases of ovarian decline, potentially aiding in fertility restoration [25]
Hematopoietic Stem Cells (HSC)	Progenitor cells supporting blood cell production (hematopoiesis); regulate immune responses; primarily sourced from bone marrow and peripheral blood [11,12,13,14,15].	-Modulate immune function, lowering inflammation [11,12,14,15]; -Secrete growth factors (HGF, SCF) that support ovarian microenvironment and cellular repair; -Support hematopoiesis and immune balance.	Valuable in treating ovarian function decline, especially when combined with MSCs for synergistic regenerative effects [25].

## Data Availability

Not applicable.

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
