# Peer review of "Dual-Double Stem Cell Ovarian Therapy: A Comprehensive Approach in Regenerative Medicine"

_ijms, 2024, doi:10.3390/ijms26010069_

Round 1
Reviewer 1 Report
Comments and Suggestions for Authors
The manuscript presents a topic of great significance to the scientific community, offering an innovative perspective supported by current and relevant references. The information is both interesting and valuable; however, the organization of the text in its present form lacks fluidity. Certain sections feel repetitive and resemble a "patchwork," with ideas and data not seamlessly integrated. Additionally, some claims need to be substantiated by stronger scientific evidence. Rather than relying on general statements such as a treatment "improves" or "worsens" fertility or other parameters, more robust data should be presented to support these assertions.
I also suggest that the text should be more superficial and should include content that covers physiological data in a comprehensive manner, in order to explain the concepts that were addressed. Diagrams demonstrating the triggering that occurs through the use of the therapies mentioned would be welcome.
Additionally, I address in the PDF of the text some suggestions.

Reviewer 2 Report
Comments and Suggestions for Authors
Comments about the manuscript:
“Dual-Double Stem Cell Ovarian Therapy: A Comprehensive Approach in Regenerative Medicine”
The use of therapy that integrates both mesenchymal stem cells (MSCs) and hematopoietic stem cells (HSCs) is being considered in particular in cases of ovarian decline or premature ovarian failure, and recent clinical applications have shown promising results. This therapy acts on tissue repair, the immune system and regeneration. MSCs then provide a microenvironment, secrete molecules promoting angiogenesis and reducing inflammation, while HSCs act on hematopoiesis and the immune system. This therapy is, however, not without risks with risks of complications (immunogenicity, tumorigenicity, immune interactions). The aim of the article proposed here is to clarify the effects of this type of ovarian therapy of the future.
This work seems to me to be a good summary of a therapeutic method of the future in the case of ovarian pathologies. If I have no criticism to make on the substance, I do however think that the form must be improved so that this manuscript can be published.
General remarks.
There are many abbreviations in the text, so it would be helpful to provide at the end or beginning of the main text, a table of abbreviations with meanings given in alphabetical order.
The division of the text deserves to be reviewed. Clearly highlight the different parts, main paragraphs, sub-paragraphs, etc. The choice of characters does not seem regular to me, which introduces a certain confusion into the text, the reading of which is not always very fluid (even though the content is very interesting and documented). I think it would be useful for authors to review the instructions for authors.
Details.
Page 3. “Dai et al. showed significant…”: Give a numbered reference.
Page 4, Clinical Research and Findings. References : according to the reference list, “Herraiz et al” are 62 and 63, not 64; “Pellicer et al” is only 63, not 62; “Tinjić et al.” is 65 ; and 66 is “Pellicer et al”. Check references and correct.
Page 6, table 1. This part starts from the table that it explains. It would be better to start from the text and give the table for illustration purposes, and not the other way around.
Page 6, table 1. Write “The table 1 summarizes…” instead of “The table 1. Summarizes…”. (delete full stop and capital letter).
Page 9, table 2. Same remark. This part starts from the table that it explains. It would be better to start from the text and give the table for illustration purposes, and not the other way around.
Page 9, table 2. Write “The table 2 summarizes…” instead of “The table 2. Summarizes…”. (delete full stop and capital letter).
Page 12. “ Herraiz et al. conducted a study… (62-64)”: Check the references (see my note above).
Page 12. “ Pellicer et al. showed significant…(62, 63)”: Check the references (see my note above).
Page 12, Comparative Molecular Mechanisms of MSCs and HSCs in Ovarian Rejuvenation. This part needs to be rewritten. Do not use a style close to telegraphic style, use sentences as at the beginning of the text.
Page 13. “ 1. Immunomodulation (Regulation of the Immune System):”: why 1 ? This part has several paragraphs 1 (1. Tissue Regeneration (Repair of Ovarian Tissue), 1. Anti-apoptotic Effects (Prevention of Cell Death…): it is therefore necessary to check, delete, rewrite.
Page 18, Risks with Autologous Non-Expanded MSC Therapy. Why is the text written in bold? Use a uniform font.
Page 19, Take home messages. This part needs to be rewritten. Do not use a style close to telegraphic style, use sentences as at the beginning of the text.
Author Response
Please see the attacment

Reviewer 3 Report
Comments and Suggestions for Authors The manuscript has several sections that are redundant and fails to provide a comprehensive overview of molecular pathways involved in loss of ovarian tissue homeostasis and molecules/pathways targeted by dual stem cell therapy for restoration of ovarian function. The authors must also write the manuscript like a review instead of listing findings from the literature. The Figure should be significantly improved upon. Hence, I recommend rejecting the manuscript. My comments for the authors are below- Dual stem cell therapy has been broadly tested and has other applications too, for example, improving cardiac function and vascular regeneration, neuroblastoma outcomes. The authors should briefly mention this in the Abstract before getting into ovarian therapy. I strongly feel that the review will benefit from adding in some schematics for the mechanism of action sections depicting molecular pathways involved in providing relief through MSC+HSC dual therapy.The authors must also discuss key ovarian carcinoma proteins and how MSC+HSC therapy targets them molecularly to restore function.
There are two conclusion sections on Page 16 which should be summarized into one. Table 1 should provide references for each function and application listed. Cell type, characteristic and function information about HSCs and MSCs provided on Pages 6-8 is redundant with introduction provided earlier and does not add anything new to the review. Similarly, the following pages (Page 9 onwards) comparing different types of MSCs and HSCs is dragged and very repetitive. All this discussion should be completed in the Introduction and more mechanistic insights need to be added to the review. The Figure provided on Page 12 should be significantly improved upon, listing molecules and pathways involved, highlighting how dual stem cell therapy targets these molecules and through secretion of which growth factors, anti-inflammatory cytokines. Pages 14, 15 - The authors must consolidate findings from the literature and stitch a meaningful summary instead of listing exosomes, microRNAs. The purpose of the review is to significantly advance understanding by providing a comprehensive overview, which I see is missing.
Author Response
Please see the attacment

Reviewer 4 Report
Comments and Suggestions for Authors
This review would summarize the latest evidence on dual-double stem cell therapy in ovarian disease. It could be a topic of great interest, but I find it really difficult to understand and/or gather the main information that I might need as a general or specific reader. The entire manuscript does not follow a logical organization, is often repetitive and never really delves into the topics.
Some examples:
At the end of Introduction, for example, a schematic figure describing the main features (from the morphology to the expression of bioactive molecules such as cytokines, growth factors, and exosomes, etc.) of both mesenchymal stem cells (MSCs) and hematopoietic stem cells (HSCs) should be helpful), especially if inserted in a diagram with their corresponding actions and targets.
Following, the section dedicated to the MSC in ovarian disease, is very superficial, never really exploring the mechanisms through which MSCs improve ovarian tissue (for example, when describing the conditioned medium (CM) derived from MSCs (page 4), any reference to the emission of exosomes/microvesicles and their potential is missing).
A summary table reporting all the actions observed in all the various studies accompanied by the respective bibliography is missing
About the paragraph : MSCs Characteristics and Mechanisms: many repetition without deepening in the description of the mechanism through which these cells help the growth by inhibiting the apoptosis
Page 7 on: The authors introduce again both kinds of cells, in a sort of list of characteristics.
Figure 1 is not clear and misleading, because not MSC or HSC are represented.
Author Response
Please see the attacment

Round 2
Reviewer 3 Report
Comments and Suggestions for Authors
The authors have satisfactorily addressed my comments and significantly improved upon the manuscript organization and presentation. I have no further questions/comments.
Reviewer 4 Report
Comments and Suggestions for Authors
The authors have significantly improved the manuscript and it is now suitable for publication.